# Neural Data Transformer 2: Multi-context Pretraining for Neural Spiking Activity

**Joel Ye**[1,2,3], **Jennifer L. Collinger**[1,3,4,5,6], **Leila Wehbe**[2,3,7], **Robert Gaunt**[1,3,4,5,6*]

[1]Rehab Neural Engineering Labs, University of Pittsburgh,
[2]Neuroscience Institute, Carnegie Mellon University,
[3]Center for the Neural Basis of Cognition, Pittsburgh,
[4]Department of Physical Medicine and Rehabilitation, University of Pittsburgh,
[5]Department of Bioengineering, University of Pittsburgh,
[6]Department of Biomedical Engineering, Carnegie Mellon University,
[7]Machine Learning Department, Carnegie Mellon University

## Abstract

The neural population spiking activity recorded by intracortical brain-computer interfaces (iBCIs) contain rich structure. Current models of such spiking activity are largely prepared for individual experimental contexts, restricting data volume to that collectable within a single session and limiting the effectiveness of deep neural networks (DNNs). The purported challenge in aggregating neural spiking data is the pervasiveness of context-dependent shifts in the neural data distributions. However, large scale unsupervised pretraining by nature spans heterogeneous data, and has proven to be a fundamental recipe for successful representation learning across deep learning. We thus develop Neural Data Transformer 2 (NDT2), a spatiotemporal Transformer for neural spiking activity, and demonstrate that pretraining can leverage motor BCI datasets that span sessions, subjects, and experimental tasks. NDT2 enables rapid adaptation to novel contexts in downstream decoding tasks and opens the path to deployment of pretrained DNNs for iBCI control. Code: `https://github.com/joel99/context_general_bci`

## 1 Introduction

Intracortical neural spiking activity contains rich statistical structure reflecting the processing it subserves. For example, motor cortical activity during reaching is characterized with low-dimensional dynamical models [1, 2], and these models can predict behavior under external perturbation and provides an interpretive lens for motor learning [3–5]. However, new models are currently prepared for each experimental context, meaning separate datasets are collected for each cortical phenomena in each subject, for each session. Meanwhile, spiking activity structure is at least somewhat stable across these contexts; for example, dominant principal components (PCs) of neural activity can remain stable across sessions, subjects, and behavioral tasks [6–9]. This structure persists in spite of turnover in recorded neurons, physiological changes in the subject, or task changes required by the experiment [10, 11]. Conserved neural population structure suggests the opportunity for models that span beyond single experimental contexts, enabling more efficient, potent analysis and application.

In this work we focus on one primary use case: neuroprosthetics powered by intracortical brain computer interfaces (iBCIs). With electrical recordings of just dozens to hundreds of channels of neuronal population spiking activity, today's iBCIs can relate this observed neural activity to behavioral intent, achieving impressive milestones such as high-speed speech decoding [12] and

---

*Correspondence to `rag53@pitt.edu`

high degree-of-freedom control of robotic arms [13]. Even so, these iBCIs currently require arduous supervised calibration in which neural activity on that day is mapped to behavioral intent. At best, cutting-edge decoders have included training data from across several days, producing thousands of trials, which is still modest by deep learning standards [12]. Single-session models still dominate the Neural Latents Benchmarks (NLB), a primary representation learning benchmark for spiking activity [14]. Thus, despite the scientifically observed conserved manifold structure, there has been little adoption of neural population models that can productively aggregate data from broader contexts.

One possible path forward is deep learning's seemingly robust recipe for leveraging heterogeneous data across domains: a generic model backbone (e.g. a Transformer [15]), unsupervised pretraining over broad data, and lightweight adaptation for a target context (e.g. through fine-tuning). The iBCI community has set the stage for this effort, for example with iBCI dataset releases (Section A.1) and Neural Data Transformer (NDT) [16], which shows Transformers, when prepared with masked autoencoding, model single-session spiking activity well. We hereafter refer to NDT as NDT1. Building on this momentum, we report that Transformer pretraining can apply to motor cortical neural spiking activity from iBCIs, and allows productive aggregation of data across contexts.

**Contributions**: Here, we contribute NDT2, a Transformer that pretrains over broad data sources of motor cortical spiking activity. NDT2 modifies NDT1 to improve scaling across heterogeneous contexts in three ways (Section 3): spatiotemporal attention, learned context embeddings, and asymmetric encode-decode [17]. We find positive transfer with data from different data sessions, subjects, and tasks, and quantify their relative value. Once pretrained, NDT2 can be rapidly tuned in novel experimental sessions. We focus on offline evaluation on motor applications, demonstrating NDT2's value in decoding unstructured monkey reaching and human iBCI cursor intent. We also show proof-of-principle real-time cursor control using NDT2 in a human with an iBCI.

## 2 Related Work

**Unsupervised neural data pretraining.** Unsupervised pretraining's broad applicability is useful in neuroscience; little is common across neural datasets except the neural data itself. Remarkably, pretraining approaches across neural data modalities are similar; of the 4 sampled in Table 1, 3 use masked autoencoding as a pretraining objective (EEG uses contrastive learning), and 3 use a Transformer backbone (ECoG uses a CNN). Still, each modality poses different challenges for pretraining; for iBCIs that record spiking activity, the primary challenge is data instability [18]. The high spatial resolution of iBCI microelectrode arrays enables recording of individual neurons and provides the signal needed for high-performance rehabilitation applications, but this fine resolution also causes high sensitivity to shifts in recording conditions [11, 18]. iBCIs typically require recalibration within hours, relative to ECoG-BCIs that may not require recalibration for days [19]. At the macroscopic end, EEG and fMRI can mostly address inter-measurement misalignment through preprocessing (e.g. registration to an atlas).

**Table 1. Neural data pretraining.** NDT2, like contemporary neural data models, are intermediate pretraining efforts, comparable to early-modern large language models like BERT [20]. Neural models vary greatly in task quality and data encoding; implanted devices (Microelectrodes, ECoG, SEEG) severely restrict the subject count available (especially with public data). Volume is estimated as full dataset size / model input size. ECoG = Electrocorticography' SEEG = Stereo electroencephalography; EEG = Electroencephalography; fMRI = Functional magnetic resonance imaging.

| Modality | Task | Estimated Pretraining Volume | Subjects |
|---|---|---|---|
| Microelectrodes (ours) | Motor reaching | 0.25M trials | ∼12 |
| ECoG: LFP [21] | Naturalistic behavior | 0.04M trials / 108 days [22] | 12 |
| SEEG: LFP [23] | Movie Viewing | 3.2M trials / 4.5K electrode-hrs | 10 |
| fMRI [24] | Varied (34 datasets) | 1.8M trials (12K scans) | 1.7K |
| EEG [25] | Clinical assessment | 0.5M trials / (26K runs [26]) | 11K |
| BERT [20] | Natural Language | 1M 'trials' (3.3B tokens) | - |

**Data aggregation for iBCI.** Data aggregation for iBCI has largely been limited to multi-session aggregation, where a session refers to an experimental period lasting up to several hours. These efforts often combine data through a method called stitching [27]. For context, the extracellular spiking signals recorded on microelectrode arrays are sometimes "sorted" by the shape of their electrical

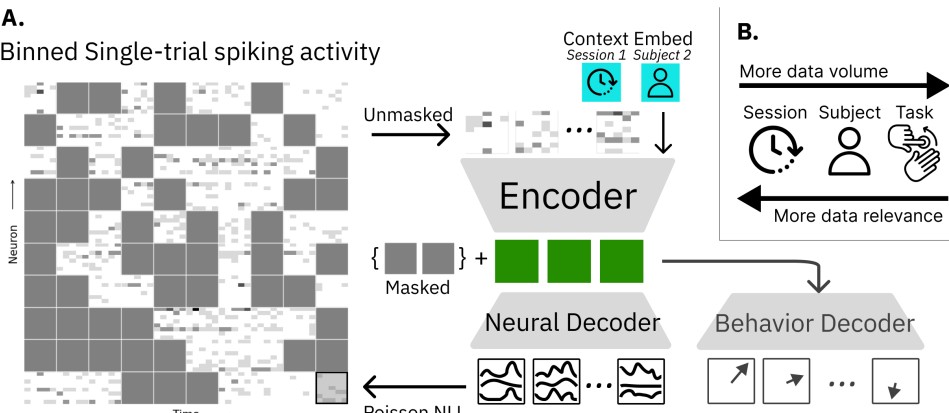

**Figure 1. A.** NDT2 is a spatiotemporal Transformer encoder-decoder in the style of He et al. [17], operating on binned spiking activity. During masked autoencoding pretraining, a spike rate decoder reconstructs masked spikes from encoder outputs; downstream, additional decoders similarly use encoder outputs for behavior prediction (e.g. cursor movement). The encoder and decoders both receive context embeddings as inputs. These embeddings are learned vectors for each unique type of metadata, such as session or subject ID. **B.** NDT2 aims to enable pretraining over diverse forms of related neural data. Neural data from other sessions within a single subject are the most relevant but limited in volume, and may be complemented by broader sources of data.

waveforms, with different shapes attributed to spikes from distinct putative neurons. Sorted data has inconsistent dimensions across sessions, but as mentioned, activity across sessions has been observed to share consistent subspace structure, as e.g. identified by PCA [7]. Stitching aims to extract this stable subspace (and also resolve neuron count differences) by learning readin and readout layers per session. Stitching has been applied to BCI and broader neuroscience applications over half a year [28–30, 11, 31]. However, even linear layers can add thousands of parameters, which risks overfitting in clinical iBCI data that comprise only dozens of trials.

Alternatively, many iBCI systems simply forgo spike sorting, which appears to have a minor performance impact [10, 18]. Then, multi-session data have consistent dimensions and can feed directly into a single model [10, 32, 33] (even if the units recorded in those dimensions shift [34]). Note that these referenced models also typically incorporate augmentation strategies centered around channel gain modulation, noising, or dropout, emphasizing robustness as a design goal.

Relative to existing efforts, NDT2 aims to aggregate a larger volume of data by considering other subjects and motor tasks (Fig. 1B). The fundamental tension we empirically study is whether the increased volume of pretraining is helpful despite the decreased relevance of heterogeneous data for any particular target context.

**Domain-adaptive vs. domain-robust decoding.** Given little training data, special approaches help enable BCI decoder use in new, shifted contexts. For example, decoders can be designed to be robust to hypothesized variability in recorded populations by promoting invariant representations through model or objective design [33, 35, 36]. Alternatively, decoders can be adapted to a novel context with further data collection, which is reasonable especially if only unsupervised neural data are required. Several approaches align data from novel contexts by learning an input mapping that minimizes distributional distance between novel encodings and pretraining encodings explicitly [11, 37, 38]. NDT2 also allows unsupervised adaptation, simply by fine-tuning without additional objectives.

## 3 Approach

### 3.1 Designing Transformers for unsupervised scaling on neural data

Transformers prepared with masked autoencoding (MAE) [20] are a competitive model for representation learning on spiking neural activity in single contexts, as measured by their performance on the NLB [14]. Cross-domain Transformer investment has also produced vital infrastructure for scaling to large datasets. Thus, we retain Transformer MAE [17], as the unsupervised pretraining recipe. This architecture is reviewed in Fig. 1A.

The primary design choice is then data tokenization, i.e. how to decompose the full iBCI spatiotemporal spiking activity into units that NDT2 will compute and predict over. Traditionally, the few hundred neurons in motor populations have been analyzed directly in terms of their population-level temporal dynamics [1]. NDT1 [16] follows this heritage and directly embeds the full population, with one token per timestep. Yet across contexts, the meaning of individual neurons may change, so operations to learn spatial representations may provide benefits. For example, Le and Shlizerman [39] and Liu et al. [36] add spatial attention to NDT's temporal attention. Yet separate space-time attention can impair performance [40] and requires padding in both space and time when training over heterogeneous data. Still, the other extreme of full neuron-wise spatial attention has prohibitive computational cost. We compromise and group $K$ neurons to a token (padding if needed), akin to pixel patches in ViTs [41]. Neural "patches" are embedded by concatenating projections of the spike counts in the patch. In pilot experiments, we find comparable efficiencies between this patching strategy and factorized attention. We opt for full attention design to easily adopt the asymmetric encoder-decoder proposed in [17]. This architecture first encodes unmasked tokens and uses these encodings to then decode masked tokens. This 2-step approach provides memory savings over the canonical uniform encoder architecture, as e.g. popularized in the language encoder, BERT [20].

We next consider token resolutions. In time, iBCI applications benefit from control rates of 50-100Hz [42]; we adopt 50Hz (20ms bins) and consider temporal context up to 2.5s (i.e. 125 tokens in time). Given a total context budget of 2K tokens (limited by GPU memory), this leaves a few dozen tokens for spatial processing. It is currently common to record from 100-200 electrodes with an interelectrode spacing of $400\mu m$ (Blackrock Utah Arrays) [29, 43], so our budget forces 16-32 channels per token. We note that future devices are likely to record from thousands of channels at a time with much higher spatial densities [44], which will warrant new spatial strategies.

NDT2 also receives learned context tokens. These tokens use known metadata to allow cheap specialization, analogous to prompt tuning [45] from language models or environment embeddings [46] from robotics. Specifically, we provide tokens reflecting session, subject, and task IDs.

## 3.2 Datasets

We pretrain models over an aggregation of datasets; the relative volume of different datasets are compared in Fig. 2A, with details in Section A.1. All datasets contains single-unit (sorted) or multi-unit (unsorted) spiking activity recorded from either monkey or human primary motor cortex (M1) during motor tasks. In particular, we focus evaluation on a publicly available monkey dataset, where the subjects performed self-paced reaching to random targets generated on a 2D screen (Random Target Task, RTT) [47], and unpublished human clinical BCI datasets. RTT data contains both sorted and unsorted activity from two monkeys over 47 sessions (∼20K seconds per monkey). RTT is ideal for assessing data scaling as it features long, near-hourly sessions of a challenging task where decoding accuracy increases with data [48]. For comparison, in the Maze NLB task, which uses cued preparation and movement periods, decoding performance saturates by 500 trials [14]. Since RTT is continuous, we split each session into 1s trials in keeping with NLB [14].

We also study M1 activity in 3 human participants with spinal cord injury (P2-P4). These participants have limited preserved motor function but can modulate neural activity in M1 using attempted movements of their own limbs. This activity can be decoded to control high degree-of-freedom behavior [13]; we restrict our study to 2D cursor control tasks to be most analogous to RTT, which also restricts targets to a 2D workspace. All experiments conducted with humans were performed under an approved Investigational Device Exemption from the FDA and were approved by the Institutional Review Board at the University of Pittsburgh. The clinical trial is registered at clinicaltrials.gov (ID: NCT01894802) and informed consent was obtained before any experimental procedures were conducted. Details on the implants and clinical trial are described in [43, 13], and similar task data are described in [49].

## 4 Results

We demonstrate the three requirements of a pretrained spiking neural data model for a BCI: a multi-context capable architecture (Section 4.1), beneficial scaled pretraining (Section 4.2) and practical deployment (Section 4.3, Section 4.4). We discuss the impact of context tokens in Section A.4.

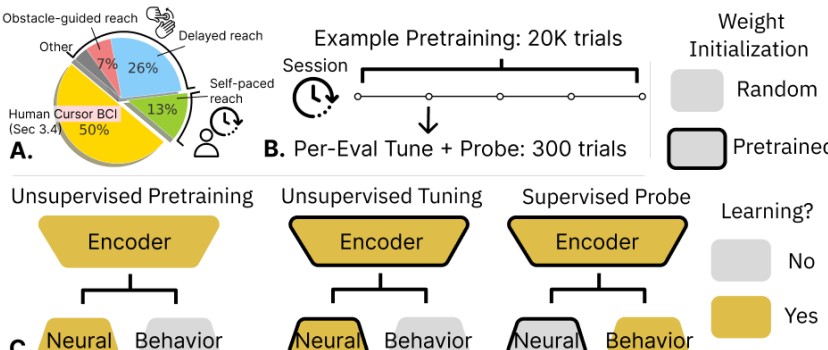

**Figure 2. Model Training: A.** We model neural activity from human and monkey reach. In monkey models, evaluation sessions are drawn from a self-paced reaching dataset [47]; multi-session and multi-subject models pretrain with other sessions in these data. The multi-task model pretrains with the other monkey reach data. Human models use a similar volume of data. **B.** A multi-session model pretrains with data from an evaluation subject, with held-out evaluation sessions. Then, for each evaluation session, we first do unsupervised tuning off the pretrained model, and then train a supervised probe off of this tuned model. **C.** We show which model components are learned (receive gradients) during pretraining and the two tuning stages. For example, supervised probes use an encoder that received both pretraining and tuning on a target session. All tuning is end to end.

**Model preparation and evaluation.** Most experiments used a 6-layer encoder ($\sim$3M parameters). NDT2 adds a 2-layer decoder (0.7M parameters) over NDT1 [16]; we ran controls to ensure this extra capacity does not confer benefits to comparison models. To ensure that our models were not bottlenecked by compute or capacity in scaling (Section 4.2), models were trained to convergence with early stopping and progressively larger models were trained until no return was observed. We pretrain with causal attention where tokens cannot attend to future timesteps, as would be the case in realtime iBCI use (though bidirectional attention improves modeling). We pretrain with $50\%$ masking and dropout of $0.1$. Further hyperparameters were not swept in general experiments; initial settings were manually tuned in pilot experiments and verified to be competitive against hyperparameter sweeps. Further training details are given in Section A.3. We briefly compare against prior reported results, but to our knowledge there is no other work that attempts similar pretraining, so we primarily compare performance within NDT-family design choices.

Model preparation and evaluation follows several steps, as summarized in Fig. 2B and C. We evaluate models on randomly drawn held-out test data from select "target" sessions (selection is later detailed per experiment). Separate models are prepared for each target session, either through fine-tuning or by training from scratch for single-context models. For unsupervised evaluation, we use the Poisson negative log-likelihood (NLL) objective, which measures reconstruction performance of randomly masked bins of test trials (Fig. 2C middle). For supervised evaluation, we report the $R^2$ of decoded kinematics, which for these experiments are 2D velocity of the reaching effector (Fig. 2C right). These supervised models are separately tuned off the per-evaluation session unsupervised model; so that all supervised scores receive the benefit of unsupervised modeling of target data [50].

## 4.1 NDT2 enables multicontext pretraining

We evaluate on five temporally spaced evaluation sessions of monkey Indy in the RTT dataset, with both sorted and unsorted processing. Both versions are important; sorted datasets contain information about spike identity and are broadly used in neuroscientific analysis, while unsorted datasets are frequently more practical in BCI applications. Velocity decoding is done by tuning all models further with a 2-layer Transformer probe (matching the reconstruction decoder). Here we provide the models with 5 minutes of data (300 training trials) for *each evaluation session*. This quantity is a good litmus test for transfer as it is sufficient to fit reasonable single-session models but remains practical for calibration. A 10% test split is used in each evaluation session (this small % is due to several sessions not containing much more than 300 trials). We pretrain models using approximately 20K trials of data, either with the remaining non-evaluation sessions of monkey Indy (Multi-Session), the sessions from the other monkey (Multi-Subject), or from non-RTT datasets entirely (Multi-Task, see Fig. 2).

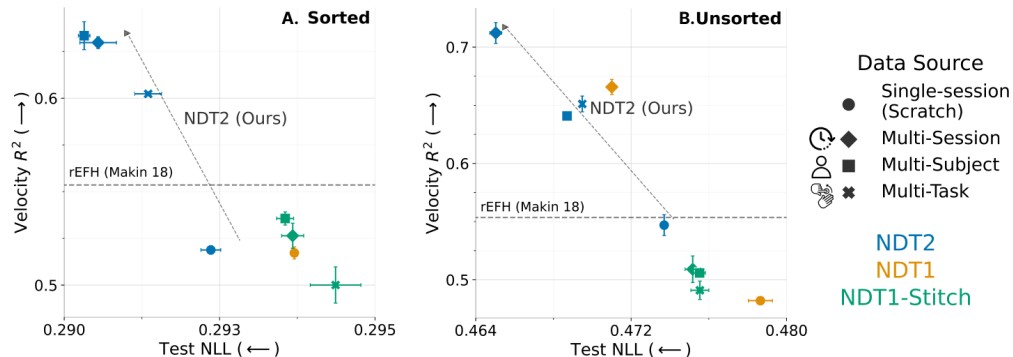

**Figure 3. NDT2 enables pretraining** over multi-session, multi-subject, and multi-task data. We show unsupervised and supervised performance (mean of 5 sessions, SEM intervals of 3 seeds) on sorted **(A)** and unsorted **(B)** spiking activity. Higher is better for $R^2$, lower is better for negative log-likelihood (NLL). Pretraining data is size-matched at 20Ks, except scratch single-session data. NDT2 improves with pretraining with all data sources, whereas stitching is ineffective. NDT1 aggregation is helpful but does not apply beyond session transfer. A reference well-tuned decoding score from the rEFH model is estimated [48].

Prior work in multi-session aggregation either use stitching layers or directly train on multi-day data with consistent unit count. Thus, we use NDT1 with stitching as a baseline for sorted data, and with or without stitching for unsorted data. NDT2 pads observed neurons in any dataset to the nearest patch multiple. All models receive identical context tokens.

We show the performance of these pretrained models for sorted and unsorted data in Fig. 3. For context, we show single-session performance achieved by NDT1 and NDT2, and the decoding performance of the nonlinear rEFH model released with the dataset [48]. rEFH scores were estimated by linearly interpolating the 16ms and 32ms scores in Makin et al. [48]. [2] Single session performance for NDT1 and NDT2 is below this baseline. However, consistent with previous findings on the advantage of spatial modeling [39], we find single-session NDT2 provides some NLL gain over NDT1. Underperforming this established baseline is not too unexpected: NDT's performance can vary widely depending on the extent of tuning (Transformers span a wide performance range on the NLB, see also Section A.3). Part of the value of pretraining is that it greatly simplifies the tuning needed for model preparation [51].

However, in sorted data shown in Fig. 3A, all pretrained NDT2 models outperform rEFH and single-session baselines, both in NLL and kinematic decoding. Surprisingly, multi-subject data work as well as multi-session data, and multi-task data provide an appreciable improvement as well. NDT-Stitch performs much worse in all cases, and in fact, cross-task pretraining brings NDT-Stitch below the single-session baseline. We expect that stitching is less useful here than in other works because other works initialize their stitching layers by exploiting stereotyped task structure across settings (see PCR in [28]), and we cannot do this because RTT does not have this structure.

The unsorted data has consistent physical meaning (electrode location) across datasets within a subject, which may particularly aid cross-session transfer. Indeed, unsorted cross-session pretraining achieves the best decoding ($> 0.7R^2$) in these experiments (Fig. 3B, blue diamond). The consistent dimensionality should not affect cross-task and cross-subject decoding, yet they also improve vs their sorted analogs, indicating the unsorted format benefits decoding in this dataset. Given this, we use unsorted formats in subsequent analysis of RTT. Otherwise, relative trends are consistent with the sorted case. Both analyses indicate that different pretraining distributions all provide some benefit for modeling a new target context, but also that not all types of pretraining data are equivalently useful.

### 4.2 NDT2 scaling across contexts

Naturally, cross-session data are likely the best type of pretraining data for its close relevance, but less relevant data can also occur in much greater volumes. To inform future pretraining data

---

[2]The rEFH model data splits vary slightly from ours: its data splits are sequential and contiguous in time, whereas we use random draws in keeping with NLB. NDT2 sorted decoding drops from 0.54 to 0.52 when mirroring Makin et al's data splits.

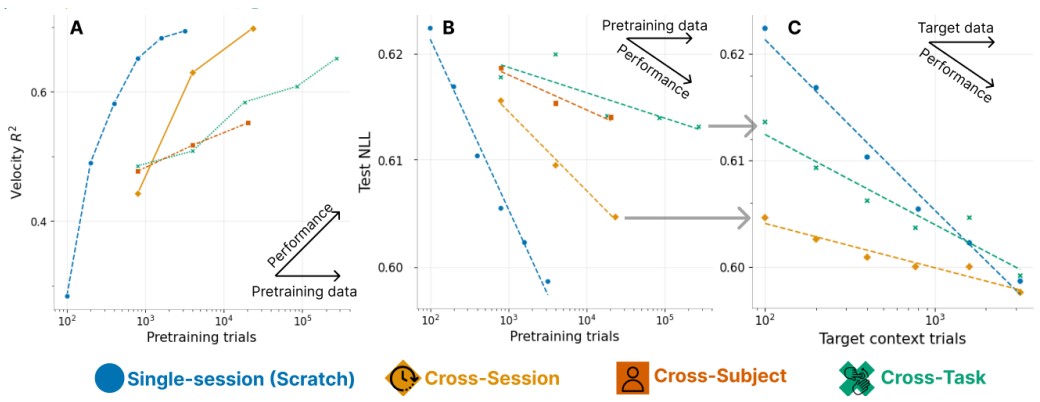

**Figure 4. Scaling of transfer on RTT.** We compare supervised $R^2$ (**A**) and unsupervised NLL scaling (**B**) as we increase the pretraining dataset size. Each point is a model; non single-session models calibrate to evaluation sessions with 100 trials. All pretraining improves over the leftmost 100-trial single-session from-scratch model, though scaling benefits vary by data sources. **C.** We seek a convergence point between pretraining and training from scratch, as we increase the number of trials we use in our target context. Models converge by 3K trials.

efforts, we perform three analyses to coarsely estimate the data affinity [52] of the three different context classes (cross-session, cross-subject, and cross-task). Previously these relationships have been grounded in shared linear subspaces [6–8]; we now quantify this in the more general generative model encompassed by DNN performance scaling.

**Scaling pretraining size.** We consider transfer as we scale pretraining size, so as to extrapolate trends that might forecast the utility of progressively larger data. Specifically, we measure performance after tuning varied pretrained models with 100 trials of calibration in a novel context. We do this for both supervised (Velocity $R^2$) and unsupervised (NLL) metrics in Fig. 4A and B respectively. To contextualize performance of cross-context scaling, we measure *in-distribution* scaling of intra-session data (Scratch). For example, the largest cross-session model tuned with 100 trials achieves a NLL similar to 1K trials of intra-session data (Fig. 4B, Cross-session vs Scratch). This shows that cross-session data can capture a practically long tail of single-session neural variance (experiments rarely exceed 1K trials), with nonsaturated benefits in pretraining dataset size. Alternately, cross-session pretraining allows decoding performance similar to the largest single-session model performance (Fig. 4A, Cross-session vs Scratch), but this scaling is starting to saturate.

Shallower slopes for cross-subject and cross-task pretraining on both metrics indicate poorer transfer to evaluation data. In the unsupervised case (Fig. 4B), cross-subject and cross-task transfer never exceed the NLL achieved with 400 single-session trials. Note that our task scaling may be pessimistic as we mix human data (Table 2) with monkey data to prepare the largest model, but the trend before this point is still shallow. However, these limitations do not clearly translate to the supervised metric (note that discrepant scaling metrics are also seen in language models [53]). For example, the decode $R^2$ achieved by the largest task-pretrained model compares with in-session models at 800 trials; the same model's NLL is most comparable that of a 300 trial single-session model.

**Convergence point with from-scratch models.** It remains unclear how rapidly pretraining benefits decrease as we increase target context data. We thus study the returns from pretraining as we vary target context calibration sizes [54] (Fig. 4C). Both models yield returns up to 3K trials, which represents about 50m of data collection in the monkey datasets, and coincidentally is the size of the largest dataset in [47]. Session pretraining provides larger gains, but task pretraining e.g. also respectably halves the data needed to achieve 0.61 NLL. This indicates pretraining is complementary to scaling target session collection efforts. This need not have been the case: even Fig. 4B suggests that task transfer by itself is ineffective at modeling the long tail of neural variance. Note that returns on supervised evaluation are likely similar or better based on Fig. 4A/B; we explore a related idea in Section 4.3.

Overall, the returns on using pretrained BCI models depends on the use case. If we are interested in best explaining neural variance, pretraining alone underperforms a moderately large in-day data collection effort (single-session model achieves lowest NLL in Fig. 4B). However, we do not see

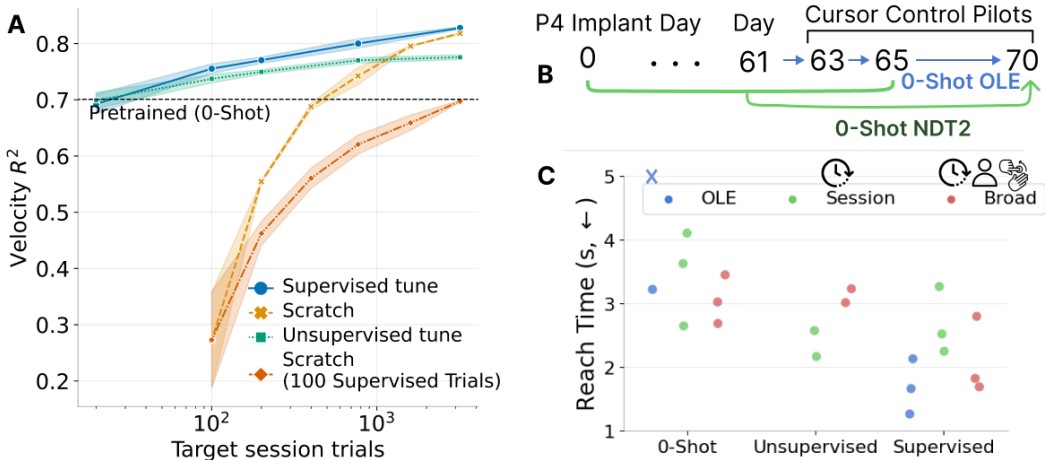

**Figure 5. Tuning and evaluating pretrained decoders. A.** In offline experiments, we compare a multisession pretrained model calibrated to a novel day against current approaches of a pretrained model without adaptation (0-Shot) and from-scratch training (yellow, orange). Both supervised and unsupervised outperforms these strategies. Standard error shown over 3 seeds. **B.** In human control pilot experiments, we evaluated models on 3 test days. "0-Shot" models use no data from test days. **C.** Average reach times in 40 center-out trials is shown for 3 decoders over 2-3 sessions. This average includes trials where target is not acquired in 10s, though this occurs 1-2 times in 40 trials. Session NDT2 uses <250 trials of data, while Broad NDT2 also includes other human and monkey data. Pretrained models provide consistent 0-Shot control while OLE [23] can sometimes fail (shown by X). Control improves with either unsupervised or supervised tuning. However, OLE appears to overtake NDT2 with supervision.

*interference* [54] in our experiments, where pretraining then tuning underperforms a from-scratch model. Thus, so long as we can afford the compute, broad pretraining is advantageous; we show these trends are repeated for two other evaluation sessions in Section A.6. We reiterate that our pretraining effort is modestly scaled; the largest pretraining only has 2 orders more data than the largest intra-context models. These conclusions may further strengthen insofar if we are able to better scale curation of pretraining data over individual experimental sessions.

### 4.3 Using NDT2 for improved decoding on novel days

**RTT Decoding**. A continuously used BCI presents the optimistic setting of having both broad unsupervised data but also multiple sessions worth of supervision for our decoder. To evaluate NDT2 in this case, we follow 1st stage unsupervised pretraining with a 2nd stage of supervised pretraining of a decoder, and finally measure the decoding performance in a novel target session in Fig. 5A. We find that given *either* supervised or unsupervised tuning in our target session, beyond smoothly improving over the 0-Shot Pretrained model's performance, achieves decoding performance on par with the best from-scratch models at all data volumes. This is true both in the realistic case where the majority of target-session data are unlabeled (100 Supervised Trials) and in the optimistic case when >1K trials of supervised data are available (Scratch). As expected, though, gains against the Scratch models are largest when target session data are limited. In sum, pretraining and fine-tuning enables practical calibration without explicit domain adaptation designs (as explored e.g. in [11, 37, 38]).

**Human BCI evaluation**. We next evaluate NDT2 in offline decoding of human motor intent, i.e. open loop trials of 2D cursor control. This shift from actual reach in monkeys is challenging: human sessions often contain few trials (e.g. 40) and intent labels are much noisier than movement recordings (intent labeling is described in Section A.1). We now evaluate a temporally contiguous experimental block, but only tune one model over this block, rather than per session, due to the high session count. We also increase test split to 50% to decrease noise from evaluating only a few trials.

We also compare broad pretraining (multi-session, subject, and task) performance (Table 2, row 1) with various data ablations. Ablating cross-subject data only has a minor performance impact (row 1 vs 2), while ablating cross-task data, which here indicates removal of data collected during online control and leaving only open-loop observation trials, hurts performance more (row 3 vs 1). We note that NDT2 fails to train using only single-sessions given extremely low trial count; we use

linear decoding performance as a single-session reference linear decoding performance. Overall, the plentiful cross-session data likely occludes further gains from cross-subject and cross-task pretraining, but note that there is no negative transfer.

Next, given reports of monkey to human transfer [55], we assess whether monkey data in either pretraining or decoder preparation improves decoding (Table 2, rows 4-7). We find that monkey data, however incorporated, reduces offline decoding performance (row 4-7 < 1). This cross-species result is the first instance of possible harm from broader pretraining, but warrants further study given the potential of transferring able-bodied monkey decoders to humans.

**Table 2. Human reach intent decoding**. We analyze how pretraining data impacts offline decoding in two people, P2 and P3. Checks (✓) indicate the data used in pretraining beyond task- and subject-specific data; human data totals 100K trials for P2 and 30K trials for P3. Intervals are SEM over 3 fine-tuning seeds. Pretraining transfers across task and somewhat across subject, but there is *no* benefit from monkey data.

| | Neural data (Unsup. pretrain) | | | Behavior (Sup. pretrain) | Velocity $R^2$ ($\uparrow$) | |
|---|---|---|---|---|---|---|
| | Cross-Subject | Cross-Task | +130K Monkey | +24K RTT (Monkey) | P2 | P3 |
| 1) | ✓ | ✓ | | | $0.503_{\pm 0.020}$ | $0.515_{\pm 0.008}$ |
| 2) | | ✓ | | | $0.487_{\pm 0.007}$ | $0.509_{\pm 0.016}$ |
| 3) | ✓ | | | | $0.444_{\pm 0.007}$ | $0.493_{\pm 0.002}$ |
| 4) | ✓ | ✓ | ✓ | ✓ | $0.486_{\pm 0.012}$ | $0.472_{\pm 0.019}$ |
| 5) | ✓ | ✓ | ✓ | | $0.490_{\pm 0.007}$ | $0.477_{\pm 0.018}$ |
| 6) | ✓ | ✓ | | ✓ | $0.474_{\pm 0.009}$ | $0.491_{\pm 0.010}$ |
| 7) | | | | ✓ | $0.443_{\pm 0.005}$ | $0.455_{\pm 0.013}$ |
| 8) | Smoothed spike ridge regression (OLE) | | | | 0.077 | 0.208 |

### 4.4 NDT2 for human cursor control pilots.

Finally, to assess NDT2's potential for deployed BCIs, we run realtime, closed-loop cursor control with one person, P4. This person was implanted recently ($\sim$ 2 months prior), and has high signal quality but limited historical data. We compare two NDT2 models, one of which pretrains with 10 minutes of participant-specific data, and one broadly pretrained on all human and monkey data, along with a baseline linear decoder (indirect OLE [56]). We evaluate the setting where decoders from recent days are available, and can either be used directly (0-shot) or updated with calibration data on the test day (Fig. 5B). All models can be supervised with test day data, but NDT2 models can also use unsupervised tuning with only neural data. In a standard center-out BCI-reaching task (methods in Section A.2), NDT2 allows consistent 0-shot use, whereas OLE sometimes fails (Fig. 5C, blue X). After either form of tuning, both NDT2 models (multisession, broad) improve. Importantly, NDT2 tunes without any additional distribution-alignment priors, as in [11, 35, 33], showing that pretraining and fine-tuning [20] may be a viable paradigm for closed loop motor BCIs.

Perhaps surprisingly, OLE provides the best control given supervised data. The performance gap and large distribution shift from offline analysis to online control is well known [18], though the specific challenge of DNN control only has basic characterization. For example, NDT2 decodes "pulsar" behavior as in [30]. Costello et al. [57] provides a possible diagnosis: these pulses reflect the ballistic reaches of the open loop training data, whereas OLE, due to its limited expressivity, will always provide continuous (but less stable) control that can be helpful in time-based metrics. Promising approaches to mitigate the pulsar failure mode include further closed-loop tuning [58] or open-loop augmentation [30]; we leave continued study of NDT2 control for future work.

## 5 Discussion

NDT2 demonstrates that broad pretraining can improve models of neural spiking activity in the motor cortex. With simple changes to the broadly used masked autoencoder Transformer, NDT2 at once spans the different distribution shifts faced in spiking data. We find distinct scaling slopes for different context classes, as opposed to a constant offset in effectiveness [59]. For assistive applications of BCI, NDT2's simple recipe for multisession aggregation is promising even if the ideal scenario of

cross-species transfer remains unclear. More broadly, we conclude that pretraining, even at a modest 10-100K trials, is useful in realistic deployment scenarios with varied levels of supervised data.

**Supervising pretrained BCI decoders.** Motor BCIs fundamentally bridge two modalities: brain data and behavior. NDT2 allows neural data pretraining, but leaves open the challenge of decoding of the full range of motor behavior. Without mapping that range, BCIs based on neural data pretraining alone will need continuous supervision. This is still a practical path forward: beyond explicit calibration phases, strategies for supervising BCIs are diverse, and can derive from user feedback [60], neural error signals [61, 62], or task-based estimation of user intent [63, 58, 64]. The ambition of pretrained *BCI* models, with broad paired coverage of the neural-behavior domain will be challenging given the experimental costs of data collection; the field of robotics suggests that scaled offline or simulated learning are important strategies given this expense. Since we lack convincing closed-loop neural data simulators (though see [65, 63]), understanding how to leverage *behavior* from existing heterogeneous datasets is an important next step.

**Negative NLB result.** NDT2 performance did not exceed current NLB SoTA on motor datasets (RTT, Maze) [66]. We first note the NLB evaluation emphasizes neural reconstruction of held-out neurons, which differs from our primary emphasis on decoding. Second, our unsupervised scaling analysis indicates that modest pretraining (100K trials) provides limited gains for neural reconstruction, especially when the target dataset has many trials (Fig. 4C), as in the NLB RTT dataset, so underperformance does not contradict our results. Revisiting the NLB after further scaling may be fruitful.

**Limitations.** NDT2's design space is under-explored. For example, we do not claim that full space-time attention is necessary over factorization. While NDT2 achieves positive transfer, further gains may come from precise mapping of context relationships [52]. Further, it is difficult to extrapolate the benefits of scaling beyond what was explored here, particularly with gains in unsupervised reconstruction appearing very limited. Our evaluation also has a limited scope, emphasizing reach-like behaviors. While these behaviors are more general than previous demonstrations of context transfer [11, 38, 33, 35], evaluating more complex behavior decoding is a practical priority. Further, NDT2 benefits are modest in human cursor control pilots, reiterating the broadly documented challenge in translating gains in offline analyses to online, human control [30, 18]. Finally, design parameters such as masking ratio may affect scaling trends, which we cannot assess due to compute limits.

**Broader Impacts.** Pretrained iBCI models may great improve iBCI usability. However, DNNs may require further safeguards to ensure that decoded behaviors, especially in real-time control scenarios, operate within reasonable safety parameters. Also, pretraining will require data from many different sources, but the landscape around human neural data privacy is still developing. While there are very few humans involved in these experiments, true deidentification remains difficult, requiring, at a minimum, consented data releases.

## Acknowledgments and Disclosure of Funding

We thank Jeff Weiss, William Hockeimer, Brian Dekleva, and Nicolas Kunigk for technical assistance in trawling human cursor experiments. We thank Cecilia Gallego-Carracedo and Patrick Marino for packaging data for this project (results do not include analysis from PM's data). We thank Joseph O'Doherty, Felix Pei, and Brianna Karpowicz for early technical advice. JY is supported by the DOE CSGF (i.e. U.S. Department of Energy, Office of Science, Office of Advanced Scientific Computing Research, Department of Energy Computational Science Graduate Fellowship under Award Number DE-SC0023112). RG consults for Blackrock Microsystems and is on the scientific advisory board of Neurowired. This research was supported in part by the University of Pittsburgh Center for Research Computing, RRID:SCR_022735, through the resources provided. Research reported in this publication was supported by the National Institute of Neurological Disorders and Stroke of the National Institutes of Health under Award Numbers R01NS121079 and UH3NS107714. The content is solely the responsibility of the authors and does not necessarily represent the official views of the National Institutes of Health.

This report was prepared as an account of work sponsored by an agency of the United States Government. Neither the United States Government nor any agency thereof, nor any of their employees, makes any warranty, express or implied, or assumes any legal liability or responsibility for the accuracy, completeness, or usefulness of any information, apparatus, product, or process

disclosed, or represents that its use would not infringe privately owned rights. Reference herein to any specific commercial product, process, or service by trade name, trademark, manufacturer, or otherwise does not necessarily constitute or imply its endorsement, recommendation, or favoring by the United States Government or any agency thereof. The views and opinions of authors expressed herein do not necessarily state or reflect those of the United States Government or any agency thereof.

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

# A  Supplementary Material

## A.1  Dataset Preparation

We aimed to perform minimal preprocessing on datasets. For pretraining, we do *not* explicitly filter for successful trial outcome as done in some neuroscientific analyses (though some datasets are released with erratic outcomes pre-filtered). Neither do we (beyond what is provided directly in datasets) filter for cross-correlated channels or low-firing neurons. We also do not z-score neuronal firing, both for simplicity and so as to not remove any potential cross-channel/session information. The one exception to this is that neurons with firing $< 0.5$Hz are removed in the sorted analysis of the O'Doherty RTT dataset, to reduce the number of spatial channels below 288. As some datasets report single unit activity and some report multi-unit activity, the dynamic range of the input data varied by an order of magnitude, with baseline firing rates varying between 0.1Hz to upwards of 50Hz. We believe additional data curation is likely to improve model quality.

In total, the max number of pretraining trials or pseudo trials was on the order of 100K trials. Each trial lasted up to 2.5s (cropped or chunked if trials were longer), and used all recorded M1 activity, and PMd activity if available.

Decoding targets were either in a standard unit or in z-scores against the dataset mean and standard deviation (not a session specific z-score). Standard units of meters/second were primarily used in most RTT analysis, except when preparing an RTT/human BCI decoder, which used respective z-scores.

**Quantifying volume** Should neural data pretraining proliferate, it will be important to distinguish pretraining efforts by neural data volume. This is tricky as the most pertinent unit of neural data seems likely to depend on the function of the neural data; we are concerned comparisons on the convenient basis of time may lead to the illusion of standardization. For example, some datasets may contain much more behavior than others in the same time periods. This issue is still present, though less severe, if we presume to only differentiate data up to orders of magnitude: compare experimental behavior vs naturalistic behavior, which may have extended periods of rest. For now, we see it reasonable to quantify the value of data in pretraining with respect to a semantic units of covariate data (experimental trials), and encourage documentation of pretraining composition.

**Reaching datasets**

- Neural Latents Benchmark motor datasets (`MC_Maze`, `MC_Maze_small`, `MC_Maze_med`, `MC_Maze_large`, `MC_RTT`): $\sim$3.7K trials.
- Churchland et al., obstacle-guided (maze) reaching, 2 monkeys, 9 sessions / $\sim$20K trials.
- Nir-Even Chen et al., delayed reaching, 2 monkeys, 12 sessions / $\sim$ 80K trials total.
- O'Doherty et al., self-paced reaching, 2 monkeys, 47 sessions / $\sim$40K seconds total.

**Isometric manipulandum datasets**

- Gallego-Carracedo et al., isometric center-out and hold, 2 monkeys, 12 sessions, $\sim$2.7K trials total.
- Dyer et al., 2 monkeys (same as above), 4 sessions/$\sim$750 trials total.

**Human BCI datasets [Private]**

- Human participant data from ongoing clinical trials. Subsetted to 2D cursor control activity, either under observation/attempted activity, partial, or full BCI control. During observation, participants observe a programmatically controlled cursors, which e.g. is performing center out at a steady pace in a trialized fashion. 130K trials. Note only observation trials are used for evaluating decoding.

## A.2  Methods for online cursor control experiments

We evaluate realtime, online cursor control using a simple Center-Out reaching task. This is a trialized task where the cursor initializes on the center of a screen and the participant reaches to 1 of 8 radially

spaced target that was cued at the start of the trial. The trial ends after the cursor is held at the target for 0.25s (preventing fly-bys from ending the trial). During open-loop calibration, the cursor is autopiloted to the target. Velocity labels are produced by assuming that the participant's intention matches the autopilot presentation. Specifically: we take the velocity of the cursor on the screen and apply a boxcar filter of 500ms to remove nonsmooth rendering, using the smoothed result as our velocity label. To span more reach conditions, we use an open-loop calibration that does not reset the cursor at the center after each trial, but instead immediately presents the next target. Our unsupervised tuning uses the neural data, but not the labels, collected in the same open loop calibration.

After each of 8 decoders (0-shot and supervised for OLE, 0-shot; supervised and unsupervised for 2 NDT models) are prepared, we evaluate each decoder in a block of 40 trials. Evaluation blocks proceed without randomization (OLE always began evaluation as it is the fastest to prepare); our results may advantage earlier decoders due to participant fatigue which accumulates over the session.

## A.3   Compute and Hyperparameter Tuning

The full, uncurated logs of all model training are available at https://wandb.ai/joelye9/context_general_bci?workspace=user-joelye9; interested users may compare specific hyperparameters logged there against those in the codebase during reproduction attempts.

**Basic hyperparameters**

1. In both pretraining and fine-tuning, we scale batch size (accumulating batches or using multi-GPU training when necessary) to be roughly proportional to full dataset so that each epoch requires 10-100 steps; we find performance is not too sensitive to batch size within an order of magnitude of this heuristic (especially in pretraining).

2. In pretraining we manually tuned LR to $5e - 4$ in initial experiments and hold it fairly constant in pretraining. We swept learning rate in our hyperparameter comparisons below.

3. In pretraining, we use learning rate warm-up for 100 epochs, and decay to $\epsilon$ by 2500 epochs. This is a high threshold that is typically not reached: training converges within 100-1K epochs for our manually tuned LR. In fine-tuning, we experimented with similar ramping schedule but settled on fixed small LR (which are typically grid-searched).

4. For RTT, we swept and found that a decoding lag of 120ms worked reasonably well. (This is similar to reports in [14]. For human BCI, we do not use decoding lag.

5. For human offline evaluation, we take the best of two evaluation hyperparameter settings: 10% or 50% masking during target-session tuning. We also report the $R^2$ only for times where the intent is non-zero; participants are not typically perfectly zero-intent during the majority of non-zero phases (i.e. data are noisy). We do not filter data by putative quality as measured by online performance in the experiment in which the data was collected; thus our calibration data includes several noisy, incomplete trials as well. Evaluation data are restricted to a contiguous set of sessions with non-trivial linear decoding.

6. Hyperparameters were not swept further during realtime pilots.

**Compute costs** We estimate computational costs with respect to data volume, as model size is held relatively static (6-12 layers, 128-384 hidden size). Most analysis was run on SLURM clusters. Pilot realtime decoding used an NVIDIA 1060/2060, where loop time was <15ms (neural data was binned at 20ms). Model tuning was performed on a remote server and took <5 minutes.

1. Fitting datasets on the order of 1K trials typically requires 20m-1hr on 12G 1080/2080-series NVIDIA GPUs.

2. 10K-20K trial datasets require 2-8 32G-V100 hours.

3. 100K+ datasets require 72 80G-A100 hours.

## A.4   NDT2 Design Notes

NDT2 introduces two primary design elements: context tokens, and patch size. We first consider the effect of context tokens. NDT2 integrates context tokens directly by adding learned tokens

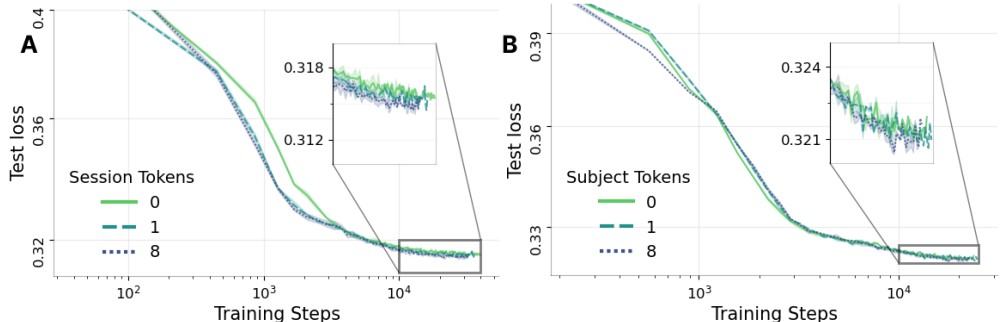

**Figure 6. Context embedding ablations. A**. Multi-session model training curves, with varying session context token count (3 seeds). Learning is improved, but additional tokens do not have notable effect. The converged score in only modestly affected. **B**. Subject transfer training curves with similarly varied token budget for subject embedding. Models receive 1 session token. There is no clear additional improvement nor harm.

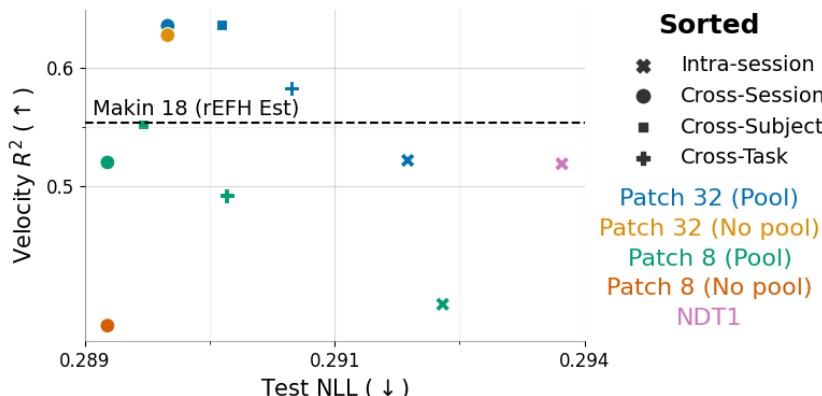

**Figure 7.** 32-neuron patches compared against 8-neuron patches. Smaller patches sometimes improve NLL but reliably decode more poorly. (Pool)-ed models average-pool all the neural data features for a given timestep before being given to the decoder, whereas regular decoders simply cross-attend all neural data features.

and adding them to the data token sequence (i.e. in-context aggregation). Our pilot experiments found no difference using cross-attention integration. The training curves of sorted multisession models augmented with context tokens, shown in Fig. 6A, demonstrate a primary effect in speeding convergence, which can be valuable in large scale pretraining. The benefit to converged NLL (some $1e-3$) is modest but non-negligible, considering the NLL resolution in Fig. 3. This trend replicates at smaller data scales (Fig. 11). Providing one session token and additionally varying the available subject tokens (Fig. 6B) has much smaller effects. However, given no harm and negligible compute overhead, we hold as a default policy to provide one token for each of session, subject, and task.

Patch size is a less intuitive hyperparameter in NDT2 than in a similar architecture in the image domain, ViT. This is because units within a patch are not meaningfully related, especially across datasets; adjacent units often correspond to different physical locations on an electrode array, and this physical meaning is typically underspecified in open data releases. Smaller patch sizes, ideally at the single neuron level, thus appear to be a better design.

However, in early experiments we explored patch size but quickly found a tradeoff. Smaller patch sizes do appear to incrementally improve neural data models, but are both more expensive computationally (to train) and statistically (to learn decoders off of). We show this in Fig. 7. Note how the unsupervised NLL is similar or better with smaller patches, but decoding is dramatically worse, regardless of whether we mean pool across the population at each timestep (Pool) or not. Smaller patches may be worth revisiting if we have a high amount of supervised data to train the decoder.

**Architectural details.** We refer readers to the codebase for full details, but provide an overview here. For each patch, NDT2 linearly projects input spike counts and concatenates these projections to form the patch embedding, zero-padding if the patch isn't full. Input spikes pass through Transformer

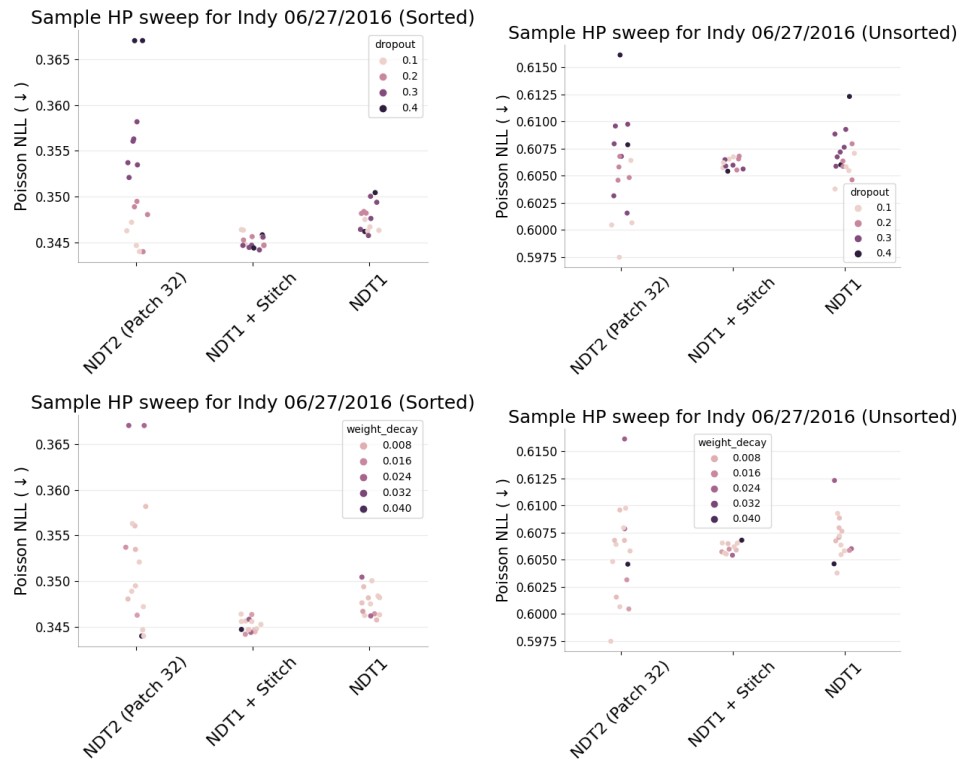

**Figure 8.** Sweeps for regularization parameters. NDT2 requires lower dropout.

encoder and decoder layers, and the final layer's output is then linearly projected to predict Poisson rates or effector velocity. NDT2 used pre-normalization layers but otherwise leave the Pytorch implementation of the Transformer layers untouched.

**HP Sweeps.**

We briefly show that NDT2 achieves higher performances than comparisons when sweeping across dropout ($[0.1, 0.4]$), weight decay ($[1e-3, 5e-2]$, and hidden size ($128, 256$). NDT2 does have higher variance, but the main sensitivity is to dropout. We run this sweep and test evaluation in one training stage, Our base NDT2 uses dropout $0.1$, hidden size $256$, weight decay $1e-2$. In the code, this experiment is configured in `exp/arch/tune_hp`, `exp/arch/tune_hp_unsort`.

### A.4.1 Mask Ratio.

We do not widely explore mask ratios due to compute constraints. In pilots throughout, we do not find that decoding is too sensitive to mask ratio (e.g. Fig. 9), but reconstruction quality is hard to compare as the inference problem depends on the masking ratio itself. The reasonable effectiveness of high mask ratios is consistent with general observations of low dimensionality and high redundancy in the code, compared to say, language [67].

### A.5 Additional exploratory experiments

**Stitching design** Our stitching implementation randomly intializes a linear readin and readout linear layer. For ease of implementation, we stitch at the output of the network encoder rather than the output of the decoder (the linear-exponential readout layer comes after per-context stitch layer). In unreported pilots, we find stitching into compressed dimensions (e.g. half of readin channels) to reduce the context-specific parameter count, or only including stitching at the readin or readout made no significant difference.

**Kinematic decoder design** There are three straightforward strategies for building a kinematic decoder with NDT2. In keeping with NLB, we could learn a linear probe on representations at each timestep, or we could use a thin Transformer decoder to allow information from multiple timesteps to aid the

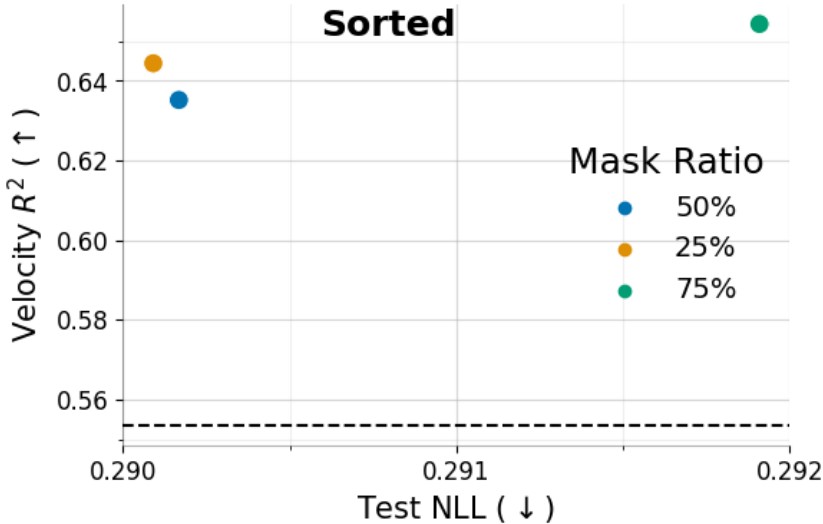

**Figure 9.** Mask ratios over 5 datasets. At test time, the given ratio is held out during unsupervised evaluation.

prediction. We experimented with both and chose the latter for minor gains. For simplicity, we run the Transformer decoder in one forward pass for all timesteps, i.e. there is no autoregressive feedback of previous kinematic estimates. We find that cross-attention for decoding queries slightly edges out in-context attention on the higher ends of the pretraining data scales we explore (e.g. 100K), and report with that setting. The primary difference is that cross-attention restricts neural data tokens from attending to the kinematic query tokens, while in-context strategies do not distinguish the two. For decoding probes, where the decoder is prepared on only a few hundred trials, we find it beneficial to mean-pool neural data tokens per timestep, and still use in-context attention (linear decoding directly works similarly).

### A.6  Single-session breakdown of experimental results

**Single-session variability** We break open the aggregate results from our primary result figures. The primary takeaways are elaborated in each caption. Overall, we note that single datasets are insufficient to make conclusions on design choices given variability in results.

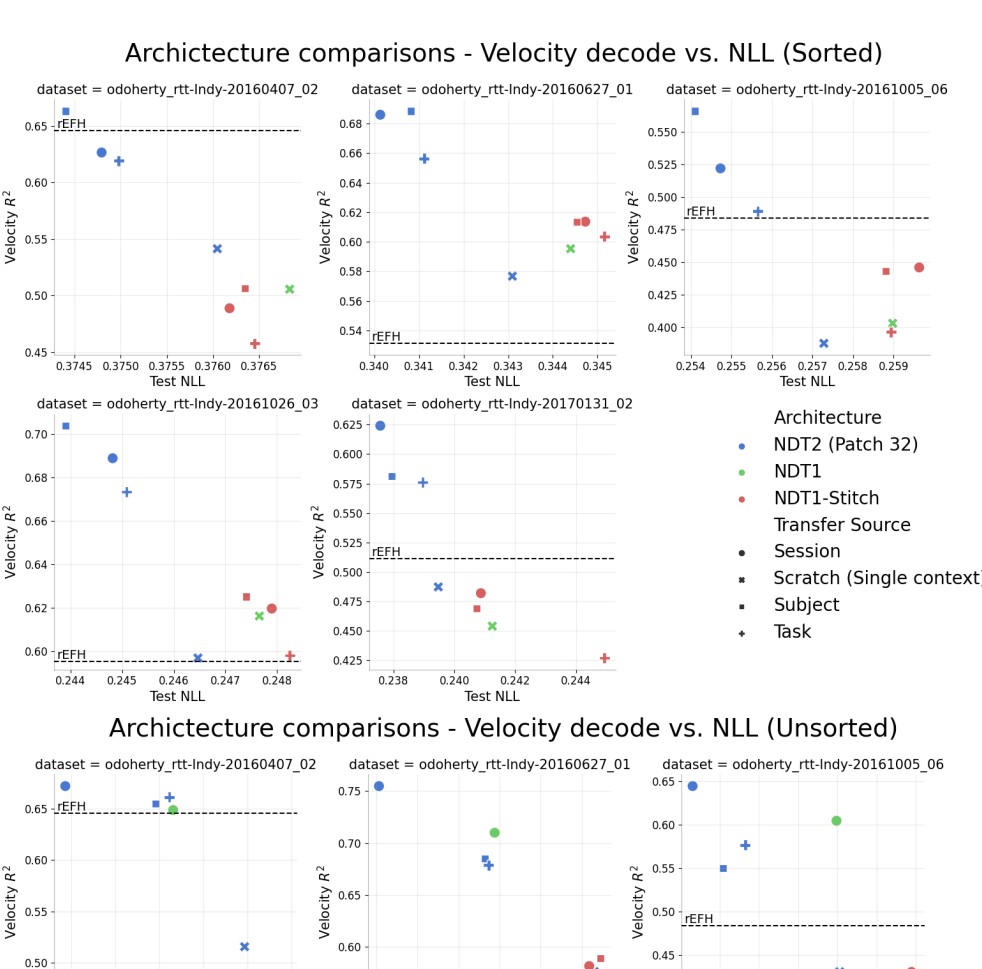

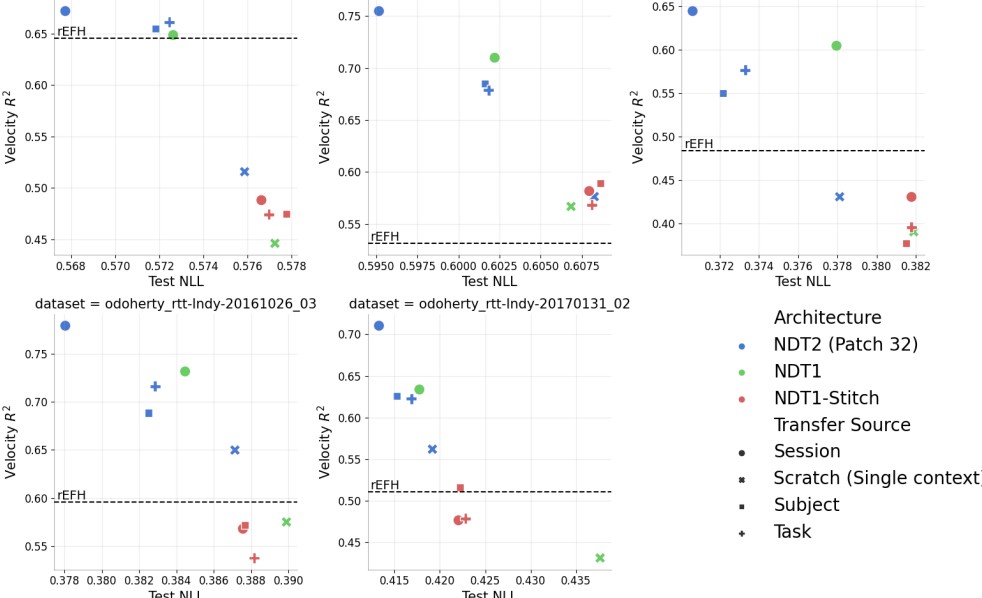

**Figure 10.** We breakout Fig. 3 into individual datasets (points indicate means on 3 seeds). NDT2 shows consistent improvements over stitching, single session baselines, and rEFH (in most cases), but the ranking between data sources shifts, particularly for decoding scores.

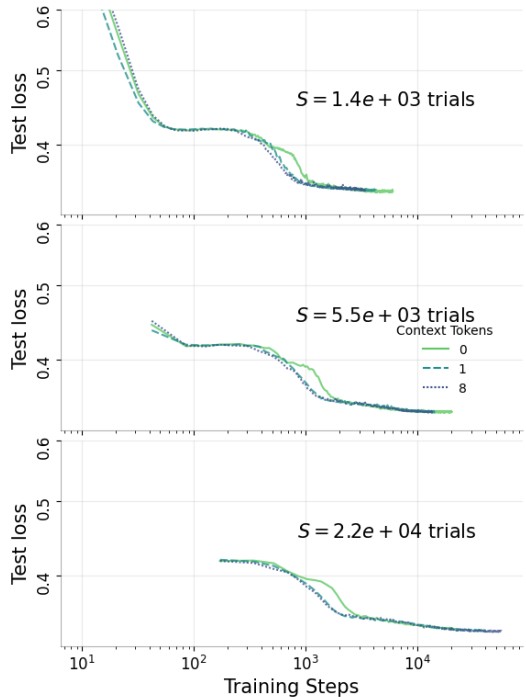

**Figure 11.** Session context tokens improve learning, as measured by unsupervised loss in model training curves, though the majority of benefit of realized with 1 token. We compare this for three data scales, annotated by S, where we scale the number of trials available per session. (The increments were 100%, 25%, and 6.25% of the data). We hypothesized that more data per session would make session tokens less relevant, but the primary effect of increasing convergence appears unchanged at these scales.

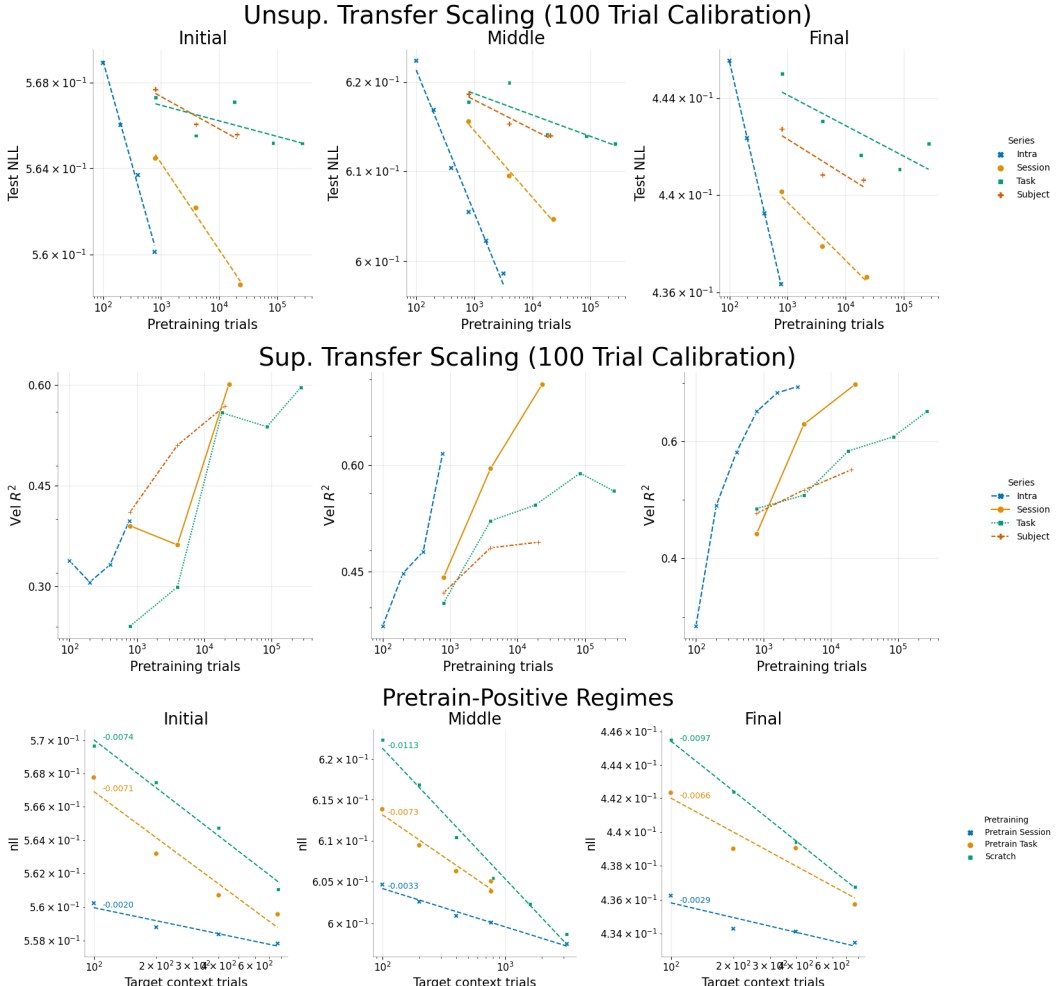

**Figure 12.** Individual dataset results for three scaling analyses. We presented the largest middle session in the primary text, here we also show results on the first and final sessions in the dataset. The unsupervised and supervised transfer scaling reiterate the previous conclusions: cross-subject and task transfer provides low returns on scaling for unsupervised reconstruction, and decoding results are much more optimistic than unsupervised results. For convergence analysis (Pretrain-positive regimes), all three trend lines suggest convergence beyond 1K trials.

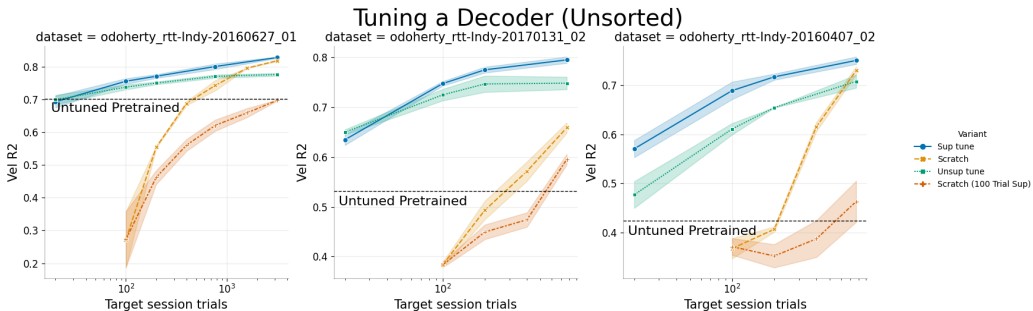

**Figure 13.** The same breakout as above for pretrained decoder tuning. Again, we find that decoder tuning reliably outperforms non-adaptive decoders.

