# OpenReview forum: "Neural Data Transformer 2: Multi-context Pretraining for Neural Spiking Activity"
_NeurIPS.cc/2023/Conference — NeurIPS 2023 poster_

### Official Review · Reviewer_mmUN · 2023-06-14

**Soundness:** 3 good
**Presentation:** 2 fair
**Contribution:** 4 excellent
**Rating:** 7
**Confidence:** 4

**Summary:**

This paper introduces the 'Neural Data Transformer 2' (NDT2), which can seamlessly integrate data from neural intracortical recordings across sessions, tasks, and subjects, by applying attention across space and time instead of having to align potentially mismatched input channels. The authors proceed to demonstrate that such cross-session/task/subject pretraining can improve decoding performance in target sessions with only little available finetuning/training data. This has clear parallels to the language literature, where large pretrained transformers have proven immensely powerful for generalizing to new contexts in zero- or few-shot settings. The authors speculate that systems neuroscience could be entering a similar era of large pretrained models, with NDT2 serving as an early proof-of-principle of the necessary cross-context generalization.

**Strengths:**

Systems neuroscientists are collecting increasingly large datasets across a wealth of brain regions, tasks, and species, and developing methods for integrating these resources remains an immensely important challenge for the field. NDT2 represents an exciting step in this direction by showing how the tools used in the vision/language literature can also drive cross-context generalization and 'few-shot' learning in systems neuroscience. As more data becomes publicly available and we improve our computational infrastructure, these ideas may help pave the way towards neural 'foundation models' that could be useful both for BCI applications and for fundamental neuroscience research.

The authors demonstrate the utility of NDT2 through a range of interesting analyses of a previously published primate reaching dataset, and they finish with an exciting new analysis of a human BCI dataset that demonstrates the potential utility of cross-context pretraining in a clinical setting.

**Weaknesses:**

The major limitation of the present study is the relatively small amount of data being used, limited to just a few monkey reaching tasks and a single human dataset. This is of course a large amount of data in the context of systems neuroscience, but it remains very small in the context of generative pre-training, and it may result in some amount of overfitting on these specific contexts (illustrated e.g. by the drop in performance on the human data when pretraining on the monkey data). One would would imagine that generative pretraining for systems neuroscience becomes much more useful when aggregating data across _many_ experiments, e.g. by scraping CRCNS. This is of course way out of scope of the present paper, which represents an exciting starting point, proof of principle, and baseline for establishing the necessary methodology as discussed above.

The presentation was not always super clear. It was at times difficult to understand what was being plotted in the figures, how the model was constructed, and how the analyses were performed (see below for a few examples). I would particularly encourage the authors to expand their figure legends to help the reader understand what is being plotted.

**Questions:**

L136: I don't understand why the authors decide to use 'trials' to refer to seconds in the RTT task since, as the authors mention, this does not correspond to actual trials. If everything is measured in units of seconds, which is a perfectly natural unit to use, why not just say seconds and minutes throughout instead of arbitrarily redefining a trial to mean a second, and then sometimes saying minutes/seconds and sometimes trials? If the authors want to provide a reference point for what happens behaviourally in a given timeframe, they could compute the median trial time (i.e. time between targets) in the RTT and report that when describing the data. For the across-task comparison, they could quantify the amount of pretraining data in other tasks in seconds as well (which is probably a fairer comparison than trials anyways).

Figure 2: it would be useful with a schematic illustrating the pretraining paradigm on the RTT data. It took me a while to figure out what the different baselines and symbols mean.

L186-187 state that "all models identically receive context tokens", but does not specify what information is contained in these tokens. Figure 3 suggests that it is some kind of session information, but even after the section specifically about context tokens, it  was unclear what information is contained in these tokens. The authors may want to clarify this somewhere.

L190-192: it is not clear that the NDT results are directly comparable to the rEFH results if the rEFH data uses contiguous training/test data and NDT does not, since any systematic drifts in electrodes/neural activity/behaviour/motivation would make the contiguous case substantially harder by requiring out-of-distribution generalization.

Figure 2: Is the 'multi-task' data from the Maze task? or does it also include other tasks? It would be worth clarifying this in the legend.

Figures 2 and 4: is it the same models being used for the supervised ($R^2$) and unsupervised (NLL) metrics?

L237-238 "This transfer is analogous, for example, to retained overlap in the first K PCs across two sessions, but generalizes nonlinearly and to many more sessions." What does this mean?

Figure 4 (very minor): I would put the panel labels on the left instead of the right of each subplot.

Figure 4: It's somewhat confusing that panels B and C use different y label notation - especially if they refer to the same quantity, which I'm not 100% sure whether they do given the inconsistent notation and minimal description in the legend.

Figure 4: It's not entirely clear what's going on in Panel C, and it may be worth clarifying in the legend. Is this panel varying the amount of within-session 'fine-tuning' data while keeping the pretraining data constant? If so, how much data was used for pre-training in each case?

Figure 2 & 4: I was surprised that pretraining on different sessions/subjects/tasks is done in an 'exclusive' manner, since a major motivation for foundation models is being able to use _more_ pretraining data. It would be interesting to see how performance changes after pretraining on either (i) just across-session data, (ii) the same amount of across-session data, but now adding across-subject data, and (iii) now also adding across-task data. Does the ability to incorporate more data benefit the model, even if the added data is in itself less 'useful'?

Figure 5: The legend feels a bit too minimal. It took a few re-readings of the main text to understand what the different lines mean.

L312 "even if the ideal scenario of cross-species transfer seems unlikely": this seems like a strong statement since transfer was attempted only with a single pre-training dataset. Cross-species transfer may still work if one were to pre-train on _many_ species to capture whatever is invariant across species rather than perhaps 'overfitting' the pretraining to monkey M1?

The above suggestions involve _adding_ many things and _removing_ nothing, which is not always easy given page limits. If space is a major limitation, it could be worthwhile to move Figure 3 and the associated text to the appendix in order to better explain the other analyses, which seem more critical to the primary message of the paper.


**Limitations:**

The authors have adequately addressed the limitations of their work.

---

> ### Author Rebuttal · Authors · 2023-08-09
>
> We are thankful for the reviewer’s enthusiastic and detailed feedback.
>
> ### Weaknesses
> We agree that an even wider pretraining is promising, but appreciate the reviewer’s recognition that NDT2 is already a substantial effort within systems neuroscience and serves as a necessary step before exploring the potential rewards of pretraining across multiple brain areas. With regards to volume, we’d like to highlight that our human models were pretrained with human data spanning over 6 years, with evaluation using data spanning over a month. This detail, along with changes for the reviewer’s other specific clarity concerns, have been added to the manuscript. We believe the added figure 2 overviewing model preparation should address the specific feedback provided, but we also provide more notes below.
>
> ### Questions
> L136: Good point; we had adopted the terminology used in the Neural Latents Benchmark, where trials were identically defined for their O’Doherty dataset. In our study, trials are used over seconds as not all time is equally behaviorally interesting; for example, in Nir-Even Chen’s delayed reaching dataset, where we might average a reach every 8s, O’Doherty’s dataset for monkey Indy averages ~1s for a reach. However, we agree this choice requires care, particularly in the context of comparing pretraining efforts in the future. While even complex natural behavior could potentially be segmented, using wall-time is simple and therefore likely the most straightforward quantify pretraining scale. We revise the manuscript to use time in broad characterizations of the dataset but trials in specific experimental comparison, and add this discussion to the appendix.
>
> Figure 2, pretraining flow, data source, model metrics: Thank you for the suggestion, we have added a schematic in the global response. In addition to the precise description requested by reviewer p9Yw, we hope the training protocols are now clear.
>
> L186, context tokens: Yes, we clarify these context tokens are simply encodings of metadata IDs, the model learns a single vector per session, subject, or task.
>
> L190: Comparison to rEFH. Indeed, we ran a simple supplementary comparison to show that NDT2 decoding score goes from 0.54 to 0.52 when mirroring the Makin data splits. We add this in a footnote. We note that Makin et al appeared to tune early stopping of their rEFH model with test data, which may advantage their model.
>
> L237-8: We agree this statement is confusing and remove it.
>
> Figure 4, Panel C: Correct, we are varying the within-session tuning data. We directly take the largest pretrained model from multi-session and multi-task data, and annotate a line from the same points in panel B to make this clear (see global response).
>
> F2 & 4, Exclusive pretraining: Indeed, we ultimately do recommend pretraining broadly on the available data. However, in practical deployment scenarios, not all types of data will be available, e.g. multisession is not available early in subject’s use. Moreover, due to data privacy concerns or practical limits, we may not generally be able to deploy multi-subject data. Thus we wanted to emphasize the relative contribution of different types of data on a small volume of target data (with the implicit understanding that already well characterized data would benefit minorly, e.g. as in 2C). The suggested inclusive, marginal return on pretraining should provide similar insight if interpreted as such, but with smaller differences between curves.
>
> L312, cross species transfer: While cross-species over many species is an exciting proposal, it might be practically challenging as the volume of relevant human or monkey data likely surpasses those of other species, except perhaps rodents. We revise our statement to make it a practical limitation rather than a theoretical one.
>
> **Overall**, we are glad the reviewer provided such rich feedback and believe the manuscript has been greatly enriched with reviewer discussion.

---

> > ### Comment · Reviewer_mmUN · 2023-08-10
> >
> > I appreciate the authors' thorough responses to all of our reviewer comments and their addition of new analyses and figures - especially the new schematics and the switch from 'trials' to 'seconds' for the RTT task.
> >
> > I look very much to forward to seeing the new exciting applications of NDT2 in the future.

---

### Official Review · Reviewer_cSw7 · 2023-07-04

**Soundness:** 2 fair
**Presentation:** 2 fair
**Contribution:** 2 fair
**Rating:** 5
**Confidence:** 4

**Summary:**

The paper introduces the NDT2 model for processing neural spiking activity data in intracortical brain-computer interfaces (iBCIs). The authors demonstrate the application and transferability of the NDT2 model across multiple experimental contexts through large-scale unsupervised pretraining. The model can aggregate neural activity data from diverse contexts and adapt rapidly to new environments. The model improves scalability in heterogeneous environments by incorporating spatiotemporal attention, learned context embeddings, and an asymmetric encode-decode structure. The paper also presents the datasets used and the evaluation results of the model.

**Strengths:**

1. Large-scale unsupervised pretraining: The NDT2 model utilizes large-scale unsupervised pretraining, enabling the aggregation of neural spiking activity data from multiple experimental contexts, overcoming the limitations of traditional models that are designed for a single context.
2. Rapid adaptation to new environments: The pretrained NDT2 model exhibits fast adaptation to new experimental environments, providing convenience for the deployment of neural brain-computer interfaces.
3. High-performance decoding: The model demonstrates excellent performance in decoding unstructured monkey reaching behavior and human iBCI cursor intent.


**Weaknesses:**

1. Computational efficiency: The full attention mechanism may increase computational costs, potentially affecting the computational efficiency in practical applications.
2. The method and background are not described in sufficient detail.


**Questions:**

1. Did the authors optimize the computational efficiency of the model to address the computational cost issue caused by the full attention mechanism?
2. Does the dataset used for evaluation have sufficient diversity to represent different real-world scenarios?


**Limitations:**

See weaknesses

---

> ### Author Rebuttal · Authors · 2023-08-09
>
> We appreciate the reviewer’s generous summary. To address reviewer cSw7 (along with the other reviewers), we have updated the manuscript with increased detail in the method and more exposition in the background (see global response) and have provided new versions of the figures to improve the clarity of the methods and the results (see rebuttal document). To address the reviewer’s specific concerns:
>
> *Computational efficiency*:
> Indeed, full attention is more expensive than other variants such as attention that has separate space and time operations. We updated our text to clarify that we compared this computationally more efficient variant in early experiments and didn’t see severe costs when using full attention. We ultimately chose to use standard full attention as we wanted a stable proof of concept of pretraining using a standard architecture. We also note that full attention computation cost does not preclude real time use of the model (see global response attachment). Nonetheless, the reviewer’s concern is important and practical. We hope future works can explore effective, efficient alternatives to full attention.
>
> *Evaluation dataset diversity*:
> The evaluation dataset is a self paced reaching task, which represents relatively naturalistic and more challenging behavior relative to the trialized, cued behavior in most other open source monkey datasets. Thus practically, we chose the most complex available open source evaluation dataset. Such data is regularly used to calibrate BCI decoders for translational control in lab environments. Identifying even more general data that applies in varied real world settings is an active research area, and we hope NDT2 and models like it can subsume this diversity once it is identified.
>
>
> We thank the reviewer for highlighting these practical concerns, and hope our responses and modifications resolve these concerns.

---

> > ### Comment · Reviewer_cSw7 · 2023-08-13
> >
> > Thank you for your response, that has effectively addressed my concerns.

---

### Official Review · Reviewer_yEFS · 2023-07-07

**Soundness:** 3 good
**Presentation:** 3 good
**Contribution:** 3 good
**Rating:** 7
**Confidence:** 3

**Summary:**

The authors present a method leveraging the original Neural Data Transformer model to span additional contexts and work across different trials and subjects. The essential claim is that pretraining using various existing datasets can improve the performance of individual iBCI decoding and control.

Update: I have updated from score from a 6 to a 7 in light of the additional work the authors provided.

**Strengths:**

Originality: While this method expands on the original NDT, it does so in an original way, using pretraining on additional datasets to include spatiotemporal attention mechanisms, learned context embeddings from diverse datasets, and asymmetric encoding and decoding methods.

Quality: The original NDT is well-studied and robust; this extension is well-motivated and uses techniques from the broader machine learning community to improve generalizability and performance in real neural datasets.

Clarity: The paper is written well but assumes details or familiarity with iBCI datasets that the NeurIPS audience may not have, or other acronyms (e.g. BERT is never defined). Much of the text is similarly dense with prerequisite knowledge.

Significance: This work has the potential to impact much of the BCI field, and future clinical applications. The dominant contribution appears to be using pretraining, which is generally known in the ML field but less so in BCI work, thus making this more significant for future endeavors.


**Weaknesses:**

The authors mention this method's use in offline decoding but it is unclear how useful this would be for practical iBCI usage.

Much of the specifics are unexplained; for example, the use of context tokens, (section starting on line 212) is described without mentioned what they are or why the authors chose to use them.

While it is clearly stated that monkey pretraining data didn't improve performance of human data, there are open questions then as to what type of pretraining data is actually useful, and how to predetermine which data is useful or not.

**Questions:**

i.	Lines 42-45 describe NDT1 and some of NDT2 but the explicit distinction between the two is not clear.
ii.	Lines 59-61 need citations.
iii.	Lines 69-71 need citations.
iv.	Line 98: additional information on the NLB task, metrics, and top model would be useful here.
v.	Line 103-104 is not clear what is meant here.
vi.	Line 161. Is this data in the supplementary somewhere?
vii.	Line 220: Figure 10 is not in the main text

**Limitations:**

The authors discuss limitations clearly in their conclusion.

Could the authors comment on line 202-203 about the lack of task structure impacting their model?

---

> ### Author Rebuttal · Authors · 2023-08-09
>
> We are excited the reviewer values the central contribution of exploring the value of pretraining in BCI. We are grateful for the specific feedback points on clarity as noted in the questions, and revised our manuscript accordingly, and address them below.
>
> > Clarity: Assumed pre-requisite knowledge
>
> We have revisited our main text and added exposition needed to introduce the previously assumed ideas from both ML and neuroscience.
>
> ### Weaknesses
> > Applications to practical iBCI usage
>
> We agree that it is critical to address how improvements in offline decoding translate to practical usage. We note that it is a substantial and active challenge to bridge this gap both technically and simply logistically, as online iBCI experiments require significant resources. See, for example, dedicated studies on deep network deployment in iBCI experiments [1, 2]. However, we are excited to add pilot online experiments (see global response) that show core benefits of pretraining and tuning: NDT2 allows decoders to work on new days, and for tuning to improve this performance.
>
> > Lack of detail on the specifics e.g. the use of context tokens.
>
> We thank the reviewer for the specific highlight, and as mentioned have revisited the methods and added detail on the specific motivation and instantiation of method details. The context tokens are learned embeddings of metadata IDs, e.g. we learn vectors for subject 1, session 3.
>
> > Open questions on determining useful data
>
> Indeed, a major contemporary challenge of pretraining, even outside of neuroscience, is identifying the right datasets to train on. In general this is currently an empirical question that is difficult to forecast a priori. We dedicated our experiments to mapping data utility at a coarse level, identifying the relative value in the categories of session, subject, and task pretraining. We hope these results provide rough guidance on potential data utility, but also look forward to future work that provides more specific recommendations.
>
>
> ### Questions
>
> i. The differences in NDT1 and NDT2 are specified in the Contributions section (l48). To reiterate: spatiotemporal attention, learned context embeddings, and asymmetric encode-decode.
>
> iv. The NLB benchmark uses similar likelihood and decoding metrics as we do. The current top model on the O’Doherty dataset is a state space model (S5) but standard Transformers and RNNs are competitive.
>
> v. We have revised the full text (~l106-114) around this spacetime attention as follows:
> ```
> …Yet factorization can impair performance [3] and requires padding in both space and time when training over heterogeneous data. However, full neuron-wise spatial attention is too costly. We thus adopt $K$-neuron patches akin to pixel patches in ViTs [4], padding data to the nearest multiple of $K$. The patch is embedded by concatenating projections of the spike counts in the patch. In pilot experiments, we find comparable efficiencies between this patching strategy and factorized attention. We opt for the former design as it enables easy adoption of the asymmetric encoder-decoder proposed in [5]....
> ```
> vi. We have rewritten to clarify: model calibration is always through fine-tuning; only single-session models (which have no broader pretraining) “pretrain” through an unsupervised step on the data in that session.
> viii. Task structure impact:Thank you, we have added clarification that stitching typically is used after an initialization procedure called principal components regression (PCR). PCR projects neural data grouped according to experimental metadata (like reaching direction), intuitively so that multiple days initialize in similar subspaces. These methods are detailed in Pandarinath et al [6]. Private correspondence suggests PCR initialization is important for more robust stitching performance.
>
> We thank the reviewer for specific feedback!
>
> [1] Deo et al. Translating deep learning to neuroprosthetic control. In bioRxiv 2023.
>
> [2] Costello et al. Balancing memorization and generalization in rnns for high performance brain-machine interfaces. In bioRxiv 2023.
>
> [3]  Arnab et al. Vivit: A video vision transformer. In CVPR 2021
>
> [4] Dosovitskiy et al. An image is worth 16x16 words: Transformers for image recognition at scale. In ICLR 2021.
>
> [5] He et al. Masked autoencoders are scalable vision learners. In CVPR 2021.
>
> [6] Pandarinath et al. Inferring single-trial neural population dynamics using sequential auto-encoders. In Nature Methods 2018. URL https://www.nature.com/articles/s41592-018-0109-9.

---

> > ### Comment · Reviewer_yEFS · 2023-08-12
> >
> > I have read over the other reviews and all the author responses. I think the updated figures are useful, in particular Figure 2 to explain what the authors mean by pretraining vs tuning vs evaluating. Rewriting the text in some locations to provide additional clarity is going to be key, as multiple reviewers were confused by the same verbiage in the paper. If the authors do that, I think the paper will be much clearer and somewhat stronger.

---

### Official Review · Reviewer_p9Yw · 2023-07-10

**Soundness:** 3 good
**Presentation:** 2 fair
**Contribution:** 3 good
**Rating:** 5
**Confidence:** 3

**Summary:**

The paper describes a transformer architecture NDT2 for spike train data. It splits a spike train in a set of 2D patches and employs a VIT with a causal auto-encoding approach to regenerate the spike-train patches. The architecture is tested on the RTT arm reaching dataset using multi-unit recordings from the primary motor cortex of a monkey and single-unit recordings from Humans with spinal cord injuries. The performance is tested in terms of negative log-likelihood and arm-velocity decoding R2. The authors report that the decoding improves when using multiple recording sessions.

**Strengths:**

The topic of BCI and decoding performance on spike recording is moving quickly and is very important for industry and neuroscience research. The paper aims to find a generic architecture that scales well using transformers which is sounds and likely to be impactful given the growing quantity of neural recordings.

The idea of using VIT is sound given the success it has had with other modalities. The dataset of random arm reaching is a well-chosen benchmark, although I was more aware of the dataset from the Shenoy lab and not this particular one.



**Weaknesses:**

My main criticism is the lack of clarity in the description of the model being chosen. I find in general that the authors have a very good overview of the existing literature but they use generic imprecise terms like "pretraining", "adaptation", "asymetric encoder-decoder", "single-sorted multi unit recording", "online iBCI decoding" which are vague if the context is not clearly defined. I would have prefered unambiguous mathematical or technical precision in ways that make it reproducible, and very precise on the technical difference from other works.

For instance here are concrete things that I would have wished to understand in more precise terms:

1) I did not understand what is meant by "an asymmetric encoder-decoder" architecture.

2) I did not understand how the NLL is computed. I assume the patches are ordered in as in VIT, suggesting that the causal masking imposes the make a prediction from previous patches in the sequence order. But within a patch, how is a decoder implementing the generation of spikes on a time-step-to-time-step basis? What is the masking during the prediction of the NLL ? Future is always masked? previous patches are masked?

3) The word pretraining is often used without a clear definition of what it means in the context. Are we talking about auto-encoder unsupervised pre-training before fine-tuning the velocity decoder?

4) I did not understand what is meant by "sorted" or "unsorted". In the context of spike trains, it was not clear to me whether we are talking about spike-sorted spike trains versus raw electrode recordings or spike-sorted spike trains with sorted unit indices, versus spike-sorted spike trains with randomly ordered unit indices.

Other sub-optimal aspects:

5) Not only it would have been better to verbalize the technical details, but there are typical metrics using the arm reaching neural recording (see for instance https://neurallatents.github.io/datasets.html) which have been standardized for comparability across models. it would have been great to be sure that the metrics are consistent and used with the same dataset.

**Questions:**

To improve this paper I recommend the authors to make the mathematical/technical description of the NDT2 more precise and unambiguous (showing equations, a pseudo code for the pre-training / evaluation algorithm, and neural network architecture with input/output tensor shapes would be a good way of doing so).

Other than the main criticisms listed above. I also wonder:

In general, the unit order in a spike train is rather arbitrary, therefore with the patch-based architecture, the group of units forming each patch could be arbitrary. Is the patch always containing units that are neighbors in anatomical space? Is that choice purely arbitrary as given by the default dataset ordering? Would that change anything?

**Limitations:**

I see no specific limitation to be reported.

---

> ### Author Rebuttal · Authors · 2023-08-09
>
> We are glad the reviewer finds the approach sound and likely to be impactful, and agree that the initial submission can be made more technically precise and broadly accessible. We thank the reviewer for their specific feedback and have incorporated all points into the manuscript; we additionally detail them here.
>
> > Lack of clarity in model description and imprecise language.
>
> We agree greater context is needed to communicate this interdisciplinary work. While descriptions of the breadth of techniques and settings in related work are necessarily imprecise, we have improved our descriptions where most relevant for our work, including the terms highlighted by the reviewer. Moreover, prompted by the reviewer’s suggestion, we add a figure overviewing the training and evaluation process, accompanied by the requested precise description. This description should complement the full code release, which provides another precise path for reproducibility.
>
> To answer the concrete questions:
> > What is an asymmetric encoder-decoder?
>
> The original NDT is a pure Transformer encoder, and processes all tokens, masked or not, in all its layers. He 21 introduces the term “asymmetric encoder-decoder.” This is a two stage architecture where only unmasked tokens are encoded in the bulk initial set of layers, and masked tokens are introduced and decoded in a final lightweight module with few layers [1]. The fact that the second stage has relatively few layers makes the model asymmetric. We have expanded this description in the text.
>
> > NLL computation and causality
>
> As in NDT1 and other neural data models, NDT2 computes NLL with a Poisson likelihood model. Specifically, predicted firing rates for each patch are compared against the binned spike count observed in the patch. For example, with a patch size of 16 channels and 20ms, the model predicts for each patch a 16-D Poisson rate vector to compare with the corresponding 16-D spike count that occurred in this 20ms. The model always uses one time bin per patch, and cannot reason about causality below the binning resolution.
>
> With regards to patch-wise causality: NDT2 attends in both space and time. Temporal attention is causal, but spatial attention is unrestricted. This contrasts with the Dosovitskiy Vision Transformer, which doesn’t restrict attention at all, and standard language models or PixelCNN, which is causal over a flat data stream. Our evaluation NLL is causal in that its computation is as done in training, where we randomly mask the input and report the average NLL on masked inputs. The data are large enough that there’s little NLL variability from masking.
>
> These are subtle points so we thank the reviewer for highlighting this and clarify the text accordingly.
>
> > Defining Pretraining
>
> Yes, our pretraining is the unsupervised masked autoencoding process, as in He 21 [1].
>
> > Sorted vs unsorted
>
> We briefly explained the sorting process on l68, but elaborate here. Microelectrode arrays record extracellular voltage signals and strong fluxes in this signal are recorded as spiking activity. Unsorted, multi-unit spiking activity is simply this activity grouped at the electrode level, so the data will have as many dimensions as there are electrodes. The sorting process further clusters this activity based on the waveform of the spikes, so sorted activity will have as many dimensions as identified spikes (which may vary across datasets).
>
> For example, a dataset with 128 channels of recording, and 2 identified neurons per electrode, would produce 256-D spike sorted data but only 128-D multi-unit activity. We do not randomize the unit indices from how they are given in the data, but these indices do not have inherent meaning.
>
> > Controlled comparison against the Neural Latents Benchmark.
>
> While the NLB is a boon for comparing neural data models, it emphasizes the evaluation of model latents over expressive decoding. As such, the NLB restricts a private set of data on which a linear probe of behavioral decoding is separately evaluated, which precludes both the evaluation of our nonlinear decoders and a controlled analysis of the impact of decoder training data. The NLB also reports spike reconstruction on _held-out_ neurons, but these held-out evaluations are not the endpoint of our brain computer interface (BCI) evaluation. Our work aims to study how different types of BCI data can scale pretrained decoder performance. Therefore, we opted for the standard unsupervised NLL reported in scaling papers in machine learning.
>
> We note that our discussion mentions our negative result on the NLB; we have clarified how the NLB setting is distinct from ours, but the negative result certainly warrants revisiting. NLB aside, we believe that a benchmark specific for BCI decoding would advance the field.
>
> > Question: Arbitrary ordering within each patch.
>
> Indeed, the spatial ordering is arbitrary in the model design, and the default of ordering them according to the datasets will place anatomically neighboring units in close patches. While we don’t believe this to affect modeling within a dataset, in early studies we did see that permuting spatial order across datasets harmed performance, suggesting that consistent patching strategies are important. However, in our supplement, A.3.2 shows that we find small patch sizes fail to decode well. We are excited for future improvements in model design that address this arbitrariness.
>
> We thank the reviewer for the opportunity to improve the clarity and reproducibility of the paper and hope that they would reconsider their assessment if our responses resolve their concerns.
>
> [1] Kaiming He, Xinlei Chen, Saining Xie, Yanghao Li, Piotr Dollár, and Ross Girshick. Masked autoencoders are scalable vision learners. Dec 2021

---

> > ### Comment · Reviewer_p9Yw · 2023-08-13
> > **Improved rating**
> >
> > Thank you for the clarifications.
> >
> > The responses to my comments are sound so I improved my rating. As I understand it most of the models details are taken from He et al. 2021. It is probably true that applying standard VIT sets a strong baseline for auto-encoding spike trains. It seems that it sets a competitive and reasonable baseline which deserves publication.
> >
> > I still encourage the authors to clarify for each of these important implementation details that they are taken from He et al. Also clarifying the jargon like saying that "(un)sorted" means spike sorted signals and not un-sorted unit channels, would be useful for the readers.

---

> > > ### Author Response · Authors · 2023-08-13
> > >
> > > Thank you for the re-evaluation and engagement! We will duly clarify these details.

---

### Author Rebuttal · Authors · 2023-08-09

We thank the reviewers for their time and specific feedback. We are pleased that the reviewers recognize that neural data pretraining is likely to be impactful (p9Yw, yEFS, mmUN), that the NDT2 approach is well-motivated yet original (p9Yw, yEFS), and that all reviewers appreciate the results achieved.

*Clarity*:
The primary criticism of all reviewers was clarity of presentation. We agree that important contextual details were omitted and that in other instances the material that was presented was difficult to follow. To address this, we have created a schematic to illustrate how different data sources were organized, how the model was prepared (p9Yw, mmUN), and also precisely describe this flow in the associated text (p9Yw). We have also updated all other figures. The first 3 figures in our attachment to this global response contains some of these updates:
- Fig 1 revises the manuscripts Fig 1, now illustrating the work’s conceptual motivation in examining the benefits of broader pretraining.
- Fig 2 is the new schematic overviewing how data was organized for model training.
- Fig 3A is a revision of our decoding figure with an elaborated legend and caption.

Further, by pooling the specific feedback provided by the reviewers, we have made substantial updates throughout the text. For example:
1. In the section on related work, we more fully describe the cited works that relate to our method (e.g. BERT, asymmetric encoder-decoder).
2. In our approach, we add concrete details to our tokenization scheme and model preparation.
3. We revise text in the description and presentation of experiments to improve clarity in tandem with revised figures.

*Open questions on data*:
Reviewers yEFS, cSw7, and mmUN raised a concern that NDT2 leaves unanswered questions about the necessity and relevance of different types of data (i.e. sessions vs. subjects vs. tasks). We had designed our experiments to provide a coarse lens to compare the value of these three major types of data, but we recognize the importance of the question and hope future works can build on our findings. Cross-species and cortical area transfer remains to be investigated, but will be challenging given the limited volume of open system neuroscience data. Nonetheless, in the words of mmUN: NDT2 represents “exciting starting point, proof of principle, and baseline for establishing the necessary methodology.”

*Practical benefits*:
Reviewers yEFS and cSw7 asked for comments on the practical benefits of the NDT2 approach. We are also very interested in this topic, and are happy to add Figure 3B in our attachment on initial human experiments, where different capabilities of NDT2 are compared. Specifically, we evaluated how pretrained models, enabled by NDT2, perform without any same-day data (zero-shot), and with both unsupervised and supervised tuning data from the evaluation day. There are acknowledged challenges in the field with offline to online translation, and while decoding performance with NDT2 after supervised tuning did not exceed current linear methods, NDT2 for the first time enabled unsupervised recalibration and also allowed reasonable zero-shot decoding performance.

We also provide individual responses to address reviewer-specific feedback.

---

### Decision · Program_Chairs · 2023-09-21

**Decision:**

Accept (poster)

**Comment:**

The paper proposes a Transformer-based architecture for modeling neural spike trains and show that pretraining on a variety of datasets improves accuracy. The reviewers appreciate the overall contributions of the paper. Their main criticism has been clarity, which the authors addressed in their rebuttal. Overall the reviewers are supportive of the paper and consider it a worthwhile contribution.